# $pMoE$: Prompting Diverse Experts Together Wins More in Visual Adaptation

**Shentong Mo**[1]**, Xufang Luo**[2]**, Dongsheng Li**[2]
[1]Carnegie Mellon University
[2]Microsoft Research

## Abstract

Parameter-efficient fine-tuning has demonstrated promising results across various visual adaptation tasks, such as classification and segmentation. Typically, prompt tuning techniques have harnessed knowledge from a single pre-trained model, whether from a general or a specialized medical domain. However, this approach typically overlooks the potential synergies that could arise from integrating diverse domain knowledge within the same tuning process. In this work, we propose a novel Mixture-of-Experts prompt tuning method called $pMoE$, which leverages the strengths of multiple expert domains through expert-specialized prompt tokens and the learnable dispatcher, effectively combining their expertise in a unified model framework. Our $pMoE$ introduces expert-specific prompt tokens and utilizes a dynamic token dispatching mechanism at various prompt layers to optimize the contribution of each domain expert during the adaptation phase. By incorporating both domain knowledge from diverse experts, the proposed $pMoE$ significantly enhances the model's versatility and applicability to a broad spectrum of tasks. We conduct extensive experiments across 47 adaptation tasks, including both classification and segmentation in general and medical domains. The results demonstrate that our $pMoE$ not only achieves superior performance with a large margin of improvements but also offers an optimal trade-off between computational efficiency and adaptation effectiveness compared to existing methods.

## 1 Introduction

The rapid advancement of unsupervised representation learning (He et al., 2020; Chen et al., 2021; Xie et al., 2021; Caron et al., 2021; Oquab et al., 2023), particularly in visual tasks, has led to an increasing demand for adaptable models that can efficiently transfer knowledge across domains. In recent years, parameter-efficient fine-tuning methods (Jia et al., 2022; Yoo et al., 2023; Mo et al., 2024b) have emerged as a powerful tool, achieving strong performance while reducing the computational burden associated with traditional fine-tuning approaches. Among these, prompt tuning, where learnable prompt tokens are added to the input sequences, has gained significant attention for its ability to adjust models pre-trained on large datasets with minimal additional parameters.

However, most existing prompt tuning approaches (Jia et al., 2022; Yoo et al., 2023; Mo et al., 2024b) focus on adapting a single pre-trained model (Chen et al., 2021; He et al., 2021), either trained on general visual tasks or specialized datasets, like medical images. While this strategy has yielded encouraging results, it inherently limits the model's capacity to benefit from cross-domain knowledge. For example, models pre-trained on general visual datasets might struggle with highly specialized medical tasks. Moreover, the capability provided by a single model is often insufficient to address a real-world downstream task to be adapted to. Solving a complex problem may require a high-level semantic understanding ability from models pre-trained with language supervision, as well as a low-level feature capturing ability from segmentation models. The challenge, therefore, is how to integrate expertise from multiple domains in a way that maximizes both performance and efficiency.

A key challenge in this context is the effective coordination of knowledge from multiple, often distinct, domain experts. Traditional approaches to fine-tuning are not designed to handle the potential conflicts

---

This work was done during Shentong Mo's internship at MSRA. Correspondence to: Xufang Luo (xufluo@microsoft.com).

or redundancies that arise when incorporating diverse sources of expertise. Furthermore, determining the optimal contribution of each expert dynamically, without inflating computational costs, remains an unsolved problem. This challenge is further compounded by the varying data characteristics and task requirements across general and specialized domains, such as the medical field, making it difficult to strike the right balance between generalization and specialization during adaptation.

To address this challenge, we propose a novel framework for prompt tuning, called *PMoE* for Mixture-of-Experts Prompt Tuning, which explicitly addresses this challenge by leveraging knowledge from multiple expert domains. Our *PMoE* first introduces expert-specific prompt tokens to each pre-trained model. Then, to facilitate information exchange and determine the contribution of each expert during adaptation, *PMoE* utilizes a learnable dispatcher module that can dynamically select and fuse tokens from diverse experts. Unlike previous methods that are confined to a single knowledge source, the proposed *PMoE* effectively integrates various domain expertise, optimizing the adaptation process across diverse tasks. Besides, the dispatcher module is compatible with all existing prompt tuning methods for a single model, and can also be extended to incorporate sophisticated architectures, making *PMoE* flexible and powerful.

We validate the effectiveness of *PMoE* through extensive experiments on 47 visual adaptation benchmarks, including both classification and segmentation tasks from general and medical domains. Our results demonstrate that *PMoE* not only achieves state-of-the-art performance, outperforming the previous method by 2.36% in terms of accuracy on ImageNet-21K classification, but also strikes a balance between computational efficiency and adaptation efficacy. We show that *PMoE* can largely improve multiple existing prompt tuning methods for a single model across all general and medical tasks. By utilizing the strengths of multiple domain experts, our approach sets a new standard for flexible and efficient visual adaptation.

Overall, our contributions can be summarized into three main folds:

- We propose *PMoE*, a novel Mixture-of-Experts Prompt Tuning framework that extends visual prompt tuning for prompting diverse experts together, allowing for effective and adaptable fine-tuning across both general and medical visual domains.

- We design a learnable dispatcher module that can flexibly select and fuse expert-specific prompt tokens, enabling dynamically allocating tokens based on the complexity and nature of the visual task.

- We conduct extensive experiments across diverse datasets, including medical image analysis and general segmentation tasks, demonstrating that pMoE significantly outperforms existing prompt-based adaptation methods.

## 2  RELATED WORK

**Visual Adaptation.** Visual adaptation seeks to transfer knowledge from pre-trained vision models to new tasks. Early approaches, such as full fine-tuning (Dosovitskiy et al., 2021), involved updating both the pre-trained backbone and task-specific heads. Recent research has focused on more efficient alternatives, particularly parameter-efficient tuning techniques. For instance, SideTune (Zhang et al., 2020a) introduced a side network that linearly interpolates between pre-trained features and side-tuned features. Bias tuning methods, such as TinyTL (Cai et al., 2020) and BitFit (Ben Zaken et al., 2022), focused on tuning only the bias terms of the backbone to reduce the number of trainable parameters. Adapter-based methods (Houlsby et al., 2019; Pfeiffer et al., 2020) injected lightweight layers into the transformer architecture, introducing task-specific parameters without retraining the full network. These approaches, however, primarily target models pre-trained in supervised settings, with fewer studies exploring parameter-efficient tuning in the context of self-supervised learning, a gap that we address with our proposed method. Our *PMoE* takes this further by introducing prompt tuning for Mixture-of-Experts to enhance adaptability and scalability across diverse visual domains.

**Visual Prompt Tuning.** Visual Prompt Tuning (VPT) (Jia et al., 2022) has recently gained traction as a parameter-efficient method for visual adaptation. VPT introduces learnable prompt tokens, appended to the input sequence, that modulate information flow through a pre-trained vision transformer. This approach has demonstrated strong performance on a variety of visual tasks, especially when used with supervised ViT backbones. Building on this, GaPT (Yoo et al., 2023) proposed adding gated prompts that control each transformer block's influence over the prompt tokens, further improving adaptability.

LSPT (Mo et al., 2024b) extended this concept by incorporating temporal prompts that retain long-term task-specific information, mitigating catastrophic forgetting. Our method expands upon these advancements by introducing Mixture-of-Experts Prompt Tuning. Unlike previous methods, our *PMoE* dynamically selects different expert prompts based on the complexity of the task, allowing for fine-grained control over model adaptation. This leads to superior performance across both general and medical domain tasks, as evidenced by our experimental results.

**Mixture-of-Experts.** Mixture-of-Experts (MoE) models, initially proposed to enhance model capacity without increasing computational cost, have shown promise in diverse areas such as natural language processing (Shazeer et al., 2017). MoE divides tasks among several "experts," each specialized in certain aspects of the input, allowing for greater task-specific specialization while maintaining parameter efficiency. Recent works in vision have begun to explore MoE for efficient transfer learning and visual adaptation (Riquelme et al., 2021). However, these approaches largely focus on architectural improvements and do not directly tackle prompt tuning in visual tasks. To our knowledge, *PMoE* is the first framework to apply MoE mechanisms to prompt tuning in vision tasks. By enabling task-dependent expert selection, *PMoE* achieves superior adaptability and performance, particularly in challenging tasks such as fine-grained classification and medical image analysis.

## 3 METHOD

Given a set of images, our target is to efficiently adapt pre-trained Vision Transformers (ViTs) to downstream visual tasks using specialized learnable prompts. We propose a novel mixture-of-experts prompt tuning framework, named *PMoE*, for capturing multiple expert blocks as prompt sources within pre-trained ViTs, as illustrated in Figure 1.

In this section, we first describe the problem setup and notations, and also revisit the visual prompt tuning technique for a single model in Section 3.1. Then, we introduce our main method, consisting of added expert prompt tokens in Section 3.2 and the dispatcher module to enable effectively utilizing knowledge from diverse experts in Section 3.3.

### 3.1 VISUAL PROMPT TUNING FOR A SINGLE MODEL

**Problem Setup.** We consider a ViT model consisting of a patch embedding layer, a stack of $L$ transformer layers, and a classification head. For an input image $\mathbf{X}$ with shape of $H \times W \times 3$, we denote the input patch tokens for the $l + 1$-th layer as $\mathbf{Z}^l = [\mathbf{z}_C^l, \mathbf{z}_1^l, ..., \mathbf{z}_N^l] \in \mathbb{R}^{(N+1) \times D}$, where $N = HW/P^2$, $P$ is the patch size, and $D$ is the dimension of the token, and $\mathbf{z}_C^l \in \mathbb{R}^{1 \times D}$ is an additional learnable classification token concatenated with patch tokens. The transformer layer processes classification and patch tokens as $\mathbf{Z}^{l+1} = \text{TransLayer}^{l+1}(\mathbf{Z}^l)$. The token $\mathbf{z}_i^0 = \text{embed}(\mathbf{x}_i)$, for $i \in \{1, 2, ..., N\}$, is obtained by embedding the $i$-th patch $\mathbf{x}_i$ of the input image $\mathbf{X}$.

**Revisit Visual Prompt Tuning.** Visual Prompt Tuning (VPT) (Jia et al., 2022) was introduced to adapt pre-trained ViTs for downstream tasks by fine-tuning continuous prompt tokens in the representation space. VPT prepends learnable prompt tokens $\mathbf{P} = [\mathbf{p}_1, ..., \mathbf{p}_{N_p}] \in \mathbb{R}^{N_p \times D}$ to the input patch tokens, where $N_p$ is the number of prompt tokens, and $D$ is the token dimension. VPT fine-tunes these prompt tokens while freezing the ViT's pre-trained weights and the classification head. VPT-deep extends this approach by injecting layer-specific prompt tokens $\mathbf{P}^l = [\mathbf{p}_1^l, ..., \mathbf{p}_{N_p}^l] \in \mathbb{R}^{N_p \times D}$ into each layer. The transformer layer processes tokens as:

$$[\mathbf{Z}_P^{l+1}, \mathbf{Z}^{l+1}] = \text{TransLayer}^{l+1}([\mathbf{P}^l, \mathbf{Z}^l]) \tag{1}$$

Here, $\mathbf{Z}_P^{l+1}$ is discarded after each block, leading to incomplete usage of accumulated prompts across layers. Our method aims to address these limitations by using these tokens for enabling prompt exchange between domain-specific experts dynamically. Some recent research improves the VPT by introducing more operations on prompt tokens (Yoo et al., 2023; Mo et al., 2024b). Note that our framework is compatible with these modifications, as long as prompt tokens exist.

### 3.2 EXPERT PROMPT TOKENS

The core of our *PMoE* lies in its ability to leverage and effectively integrate multiple domain-specific experts[1] through a set of learnable prompt tokens. Thus, we first introduce Expert Prompt Tokens

---

[1] In this paper, an expert refers to a pre-trained model.

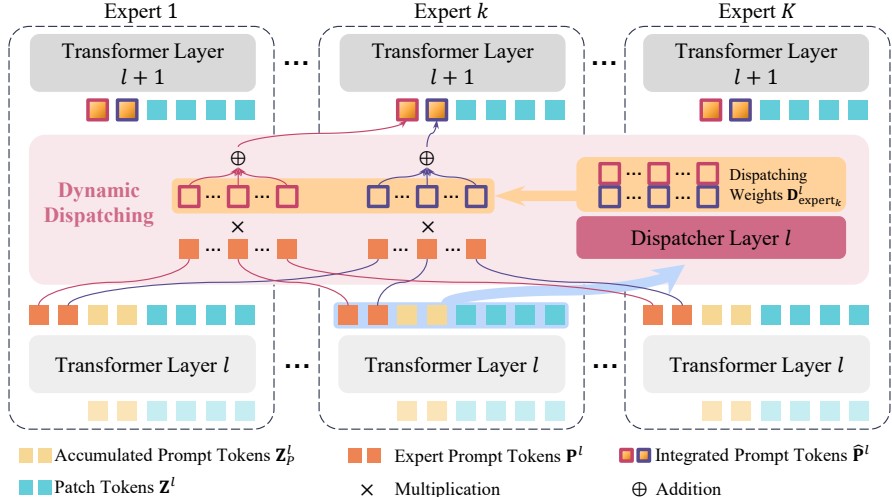

Figure 1: Illustration of the proposed *pMoE* framework. Here, we demonstrate how the dispatcher handles tokens and produces integrated prompt tokens for Expert $k$, with the same method applied to other experts as well. The dynamic dispatching method takes expert prompt tokens from all experts and the state of the current expert as inputs, and outputs dispatching weights for controlling portions to integrate prompt tokens for the next layer. Different colors represent distinct weight groups, applied to corresponding expert prompt tokens, yielding different integrated prompt tokens. This dynamic dispatching mechanism ensures communication and interaction among diverse experts, making the model contribute the most relevant knowledge to the final output.

(EPTs), specialized to capture domain-specific knowledge, allowing the model to access diverse expert knowledge dynamically during the adaptation process. Specifically, for each domain expert, a dedicated set of learnable prompt tokens is assigned. These expert-specific tokens are injected at the input layer, similar to VPT, but with multiple expert sources. The prompt tokens can be defined as:

$$\mathbf{P} = \{\mathbf{P}_{\text{expert}_1}, \dots, \mathbf{P}_{\text{expert}_k}, \dots, \mathbf{P}_{\text{expert}_K}\} \tag{2}$$

where $\mathbf{P}_{\text{expert}_k}$ represents prompt tokens for the $k$-th domain expert[2]. For instance, $\mathbf{P}_{\text{expert}_1}$ can be injected to the DINO (Oquab et al., 2023) model to capture some basic discriminative features, and injecting $\mathbf{P}_{\text{expert}_2}$ to a medical vision encoder (e.g., LVM-Med (Nguyen et al., 2023)) with richer domain knowledge can provide deeper understanding on medical images. Moreover, similar to VPT-deep, EPTs can also be injected layer-wisely, catching different information in different layers.

### 3.3 Dynamically Select and Fuse Tokens with the Dispatcher

After obtaining domain-specific knowledge, we consider the deigns of utilizing EPTs and patch tokens, for effectively leveraging knowledge from multiple domain experts and contributing to better performance in the downstream task than using any of the single model. To address the challenge of handling potential conflicts or redundancies among diverse experts, and identifying key information for downstream tasks, the framework must (1) enable communications among experts to facilitate information exchanging and then (2) decide what information should be preserved accordingly. Therefore, *pMoE* introduces a dynamic dispatcher module that takes EPTs from all experts, and decide what should be preserved according to the state of the current expert.

**Inserting Dispatcher Layer.** A learnable dispatcher layer (DispLayer$^l$) can be inserted across all experts, before tokens entering the subsequent transformer layer (index $l + 1$) of every expert. Although the dispatcher layer is shared for all experts for efficiency and facilitating communications, it needs to wisely choose the most relevant information for each expert, aggregating them into Integrated Prompt Tokens (IPTs) for further processing. This is enabled with a dynamic dispatching method (explained in the next paragraph). Then, IPTs ($\hat{\mathbf{P}}^l_{\text{expert}_k}$ for expert $k$) and patch tokens $\mathbf{Z}^l_{\text{expert}_k}$ are sent

---

[2] We use notations with subscript "expert" and its index to denote expert-specific information, while notions without this subscript represent the aggregation of information from all experts.

to $\text{TransLayer}^{l+1}_{\text{expert}_k}$, producing accumulated prompt tokens $\mathbf{Z}^{l+1}_{P,\text{expert}_k}$ and patch tokens $\mathbf{Z}^{l+1}_{\text{expert}_k}$.

$$[\mathbf{Z}^{l+1}_{P,\text{expert}_k}, \mathbf{Z}^{l+1}_{\text{expert}_k}] = \text{TransLayer}^{l+1}_{\text{expert}_k}([\hat{\mathbf{P}}^l_{\text{expert}_k}, \mathbf{Z}^l_{\text{expert}_k}]) \tag{3}$$

**Dynamic Dispatching Weights.** The dynamic dispatching method should enable communication among all experts and also consider expert-specific states. It contains two steps. (1) The dispatcher layer takes expert-specific states as inputs, thus making different decisions for each expert accordingly, and output dynamic dispatching weights $\mathbf{D}^l_{\text{expert}_k}$. Expert-specific states contain EPTs of the current expert $\mathbf{P}^l_{\text{expert}_k}$, accumulated prompt tokens from the last layer $\mathbf{Z}^l_{P,\text{expert}_k}$, and patch tokens from the last layer $\mathbf{Z}^l_{\text{expert}_k}$. (2) IPTs $\hat{\mathbf{P}}^l_{\text{expert}_k}$ are obtained by weighted addition of all EPTs using expert-specific weights $\mathbf{D}^l_{\text{expert}_k}$. This process can be expressed as:

$$\mathbf{D}^l_{\text{expert}_k} = \text{DispLayer}^l([\mathbf{P}^l_{\text{expert}_k}, \mathbf{Z}^l_{P,\text{expert}_k}, \mathbf{Z}^l_{\text{expert}_k}])$$
$$\hat{\mathbf{P}}^l_{\text{expert}_k}[n] = \mathbf{D}^l_{\text{expert}_k}[n] \times \mathbf{P}^l[\cdot, n],$$
$$\hat{\mathbf{P}}^l_{\text{expert}_k} = [\hat{\mathbf{P}}^l_{\text{expert}_k}[1], \ldots, \hat{\mathbf{P}}^l_{\text{expert}_k}[n], \ldots, \hat{\mathbf{P}}^l_{\text{expert}_k}[N_p]].$$

Let $N_p$ be the number of prompt tokens for each expert and $K$ be the number of experts. $D$ is the dimension of the token. Here, $\mathbf{D}^l_{\text{expert}_k} \in \mathbb{R}^{N_p \times K}$ and $\mathbf{D}^l_{\text{expert}_k}[n] \in \mathbb{R}^{1 \times K}$ where $n$ is the index. $\mathbf{P}^l \in \mathbb{R}^{K \times N_p \times D}$ and we use $\mathbf{P}^l[\cdot, n] \in \mathbb{R}^{K \times D}$ to denote the $n$-th EPT of all experts. So, the $n$-th IPT of the current expert is the weighted combination of $n$-th EPTs of all experts. The $\text{DispLayer}^l$ converts tokens with dimension $D$ into weights with dimension $K$ via following operations:

$$[\mathbf{D}^l_{E,\text{expert}_k}; \mathbf{D}^l_{P,\text{expert}_k}; \mathbf{D}^l_{Z,\text{expert}_k}] = \text{MLPs}([\mathbf{P}^l_{\text{expert}_k}; \mathbf{Z}^l_{P,\text{expert}_k}; \mathbf{Z}^l_{\text{expert}_k}]),$$
$$\mathbf{D}^l_{\text{expert}_k} = \text{softmax}(\mathbf{D}^l_{E,\text{expert}_k} + \mathbf{D}^l_{P,\text{expert}_k} + \text{Average}(\mathbf{D}^l_{Z,\text{expert}_k})).$$

where $\mathbf{P}^l_{\text{expert}_k}, \mathbf{Z}^l_{P,\text{expert}_k} \in \mathbb{R}^{N_p \times D}$, $\mathbf{D}^l_{E,\text{expert}_k}, \mathbf{D}^l_{P,\text{expert}_k} \in \mathbb{R}^{N_p \times K}$, and $\mathbf{Z}^l_{\text{expert}_k} \in \mathbb{R}^{N_z \times D}$, $\mathbf{D}^l_{Z,\text{expert}_k} \in \mathbb{R}^{N_z \times K}$, where $N_z$ denote the number of patch tokens. We take the average over patches and then broadcast it to $N_p$. And softmax normalization is performed over the second dimension. Detailed algorithm is in Appendix B.

In summary, the dispatcher takes all EPTs with the learned domain knowledge inside them, and patch tokens of the current expert, and then send integrated tokens to the next layer of the current expert. This comprehensive consideration ensures communication and interaction among diverse experts. Besides, the dispatching decision is learned by the dispatcher layer, which considers both information in all experts and the state of the current expert, making each expert contributes necessary knowledge to the final output accordingly.

**Explanations and Remarks.** The dynamic dispatching mechanism in *PMoE* *maintains the computational efficiency benefits of prompt tuning methods*. Since the dispatcher layer comprehensively accounts for information utilization and is shared across all experts, it remains lightweight, ensuring that *PMoE* does not largely increase tunable parameters — most still come from the original prompt tokens. Our results (ref. Section C) also demonstrate that when the same number of tunable prompt tokens is introduced across all models, *PMoE* can achieve a synergistic effect, where combined performance exceeds individual contributions. This strategy enhances adaptation performance while minimizing the risk of overfitting to a single domain.

In *PMoE*, IPTs are designed to interact with patch tokens only from the corresponding expert in the transformer layer, but EPTs can communicate and exchange across all experts in the dispatcher layer. This is different with naively using shared prompts for all experts, enabling learning expert-specific knowledge and collaboration among all experts at the same time.

Note that *the dispatcher module is flexible and can be added only to the first layer, similar to vanilla VPT*. Here, EPTs interact with the input image patches, and their contribution is dynamically weighted based on the task and data characteristics. When the dispatcher is added to other layers, it selectively decides the expert-specific prompts that should propagate through each layer. This ensures that knowledge from diverse experts can be exploited optimally across all stages of adaptation.

Besides, tokens $\mathbf{Z}^l_P$ originally discarded by VPT-deep in equation 1 are well considered in *PMoE*, *addressing the issue of incomplete usage of accumulated prompts across layers*.

Table 1: Comparison results of visual prompt tuning of supervised ImageNet-21K ViT-B/16 weights on VTAB-1K benchmarks. Numbers in (·) denote the number of downstream datasets.

| Method | Natural (7) | Specialized (4) | Structured (8) | Average |
|---|---|---|---|---|
| VPT (Jia et al., 2022) | 78.48 | 82.43 | 54.98 | 69.42 |
| EXPRES (Das et al., 2023) | 79.70 | 84.00 | 55.00 | 70.21 |
| SNF (Wang et al., 2023) | 83.79 | 86.13 | 59.61 | 74.10 |
| Bi-AdaptFormer (Jie et al., 2023) | 82.11 | 86.40 | 62.43 | 74.73 |
| LSPT (Mo et al., 2024b) | 85.26 | 88.57 | 66.25 | 77.95 |
| LSPT + pMoE (ours) | **87.18** | **90.25** | **69.32** | **80.31** |

## 4 EXPERIMENTS

### 4.1 EXPERIMENTAL SETUP

**Datasets.** For the general domain, we leverage two popular classification benchmarks: FGVC (Wah et al., 2011; Nilsback & Zisserman, 2008; Gebru et al., 2017; Khosla et al., 2011; Van Horn et al., 2015) and VTAB-1K (Zhai et al., 2019). We follow the same training and validation split as prior work (Jia et al., 2022; Yoo et al., 2023; Mo et al., 2024b). For medical imaging, we utilize a broad set of datasets from the Med-VTAB benchmark (Mo et al., 2024a), covering a wide range of medical tasks. For the segmentation tasks, we include the ADE20K (Zhou et al., 2017; 2018), Kvasir-seg polyp (Jha et al., 2020) and the ISIC skin lesion dataset (Gutman et al., 2016). Both datasets are evaluated using 5-fold cross-validation, with performance reported as the average across test splits.

**Evaluation Metrics.** To evaluate the performance of $pMoE$ across general and medical image analysis tasks, we employ several standard evaluation metrics. For classification tasks, we use classification accuracy and the area under the receiver operating characteristic curve (AUC). For segmentation tasks, we assess performance using mask Intersection-over-Union (mIoU). These metrics provide a comprehensive measure of our model's ability to generalize across both general image classification and medical image analysis tasks, ensuring an accurate and fair comparison with existing methods.

**Implementation.** We implement $pMoE$ using PyTorch (Paszke et al., 2019) library. The pre-trained Vision Transformers (ViTs) used in our experiments are initialized from publicly available weights of ViT-B/16 (Dosovitskiy et al., 2021) models. For all experiments, we freeze the backbone transformer layers and fine-tune only the newly introduced prompt tokens and layers. For both general and medical datasets, we fine-tune the prompt tokens with the AdamW optimizer (Loshchilov & Hutter, 2017), using a learning rate of 1e-4 and a weight decay of 1e-5. The batch size is set to 32 for all datasets, and training is conducted over 30 epochs.

### 4.2 COMPARISON TO PRIOR WORK

To comprehensively evaluate the performance of $pMoE$, we conducted extensive comparisons against state-of-the-art adaptation techniques across both general and medical domain tasks. These experiments were designed to test classification and segmentation benchmarks, offering a holistic view of how well $pMoE$ adapts in varied environments.

**Supervised ImageNet-21k Training.** To validate the effectiveness of $pMoE$ in visual prompt tuning under supervised settings, we evaluated its performance using the ImageNet-21K pre-trained ViT-B/16 model on the VTAB-1K benchmarks (Zhai et al., 2019). We compared $pMoE$ with several recent methods (Das et al., 2023; Wang et al., 2023; Jie et al., 2023), including VPT (Jia et al., 2022), EXPRES (Das et al., 2023), SNF (Wang et al., 2023), Bi-AdaptFormer (Jie et al., 2023), and LSPT (Mo et al., 2024b), all of which are leading methods in visual prompt tuning. As shown in Table 1, $pMoE$ achieves the highest average score across the VTAB-1K benchmarks, surpassing all previous methods in the Natural, Specialized, and Structured task categories. Specifically, $pMoE$ outperforms LSPT (Mo et al., 2024b) by 1.92 on Natural tasks, 1.68 on Specialized tasks, and 3.07 on Structured tasks. The superior performance of $pMoE$ is attributed to its dynamic prompt token mechanism, which effectively captures diverse domain-specific knowledge. Additionally, our approach significantly improves upon EXPRES (Das et al., 2023), which focuses on learning residual tokens, by leveraging a more flexible prompt architecture that dynamically integrates multiple expert domains. These results confirm the efficacy of $pMoE$ in achieving state-of-the-art results for visual prompt tuning, both in terms of accuracy and adaptability, across diverse image classification.

Table 2: Comparison results of DINO v2 pre-trained vision transformers on FGVC and VTAB-1k datasets. Total Params denotes the total number of parameters for the backbone encoder ViT-B, prompt tokens or adapter parameters, and the task heads.

| Method | Total Params | CUB | Flowers | Cars | Dogs | NABirds | Nature | Specialized | Structured |
|---|---|---|---|---|---|---|---|---|---|
| VPT (Jia et al., 2022) | 1.02X | 82.67 | 94.41 | 79.18 | 83.33 | 75.99 | 70.27 | 83.04 | 42.38 |
| VPT (Jia et al., 2022) | 1.04X | 82.95 | 94.65 | 79.37 | 83.62 | 76.21 | 70.55 | 83.42 | 42.69 |
| VPT + *PMoE* (ours) | 1.04X | **83.38** | **94.85** | **79.72** | **83.95** | **76.73** | **71.03** | **83.85** | **43.11** |
| GaPT (Yoo et al., 2023) | 1.02X | 82.86 | 93.71 | 79.02 | 83.37 | 76.02 | 74.84 | 83.38 | 49.10 |
| GaPT (Yoo et al., 2023) | 1.04X | 83.05 | 93.98 | 79.25 | 83.68 | 76.28 | 75.13 | 83.67 | 49.46 |
| GaPT + *PMoE* (ours) | 1.04X | **83.52** | **94.62** | **79.69** | **84.06** | **76.85** | **75.68** | **84.15** | **50.02** |
| LSPT (Mo et al., 2024b) | 1.05X | 84.29 | 95.06 | 80.12 | 84.25 | 77.16 | 77.19 | 85.69 | 52.82 |
| LSPT (Mo et al., 2024b) | 1.10X | 84.85 | 95.57 | 80.63 | 84.53 | 77.52 | 78.13 | 86.35 | 53.38 |
| LSPT + *PMoE* (ours) | 1.10X | **86.07** | **96.58** | **81.12** | **85.17** | **78.06** | **78.82** | **86.98** | **53.97** |

Table 3: Comparison results of visual prompt tuning of DINO v2 pre-trained vision transformers on color images. Total Params denotes the total number of parameters for the backbone encoder ViT-B, prompt tokens or adapter parameters, and the task heads.

| Method | Total Params | Hyper Polyp | MESAD Prosta | AMLC Cell | APTOS Eye | ISIC Skin | Kvasir Polyp | LCBC Cell | MLLB Cell | EyePACS Eye |
|---|---|---|---|---|---|---|---|---|---|---|
| VPT (Jia et al., 2022) | 1.02X | 62.89 | 43.78 | 35.75 | 57.52 | 50.89 | 66.53 | 42.87 | 35.37 | 48.75 |
| VPT (Jia et al., 2022) | 1.04X | 63.15 | 43.97 | 35.98 | 57.83 | 51.06 | 66.92 | 43.11 | 35.76 | 48.97 |
| VPT + *PMoE* (ours) | 1.04X | **65.23** | **45.08** | **37.53** | **59.62** | **52.75** | **68.73** | **45.68** | **37.89** | **50.67** |
| GaPT (Yoo et al., 2023) | 1.02X | 65.18 | 45.79 | 37.26 | 59.37 | 51.58 | 67.13 | 45.16 | 36.85 | 51.57 |
| GaPT (Yoo et al., 2023) | 1.04X | 65.57 | 46.08 | 37.53 | 59.68 | 51.82 | 67.46 | 45.52 | 37.21 | 51.92 |
| GaPT + *PMoE* (ours) | 1.04X | **67.43** | **47.97** | **39.45** | **61.27** | **53.56** | **69.82** | **46.73** | **39.65** | **53.78** |
| LSPT (Mo et al., 2024b) | 1.05X | 67.23 | 47.53 | 38.72 | 61.25 | 53.62 | 69.79 | 47.51 | 38.92 | 52.86 |
| LSPT (Mo et al., 2024b) | 1.10X | 67.95 | 48.16 | 39.38 | 61.87 | 54.37 | 71.53 | 48.25 | 39.67 | 53.58 |
| LSPT + *PMoE* (ours) | 1.10X | **69.83** | **50.26** | **41.25** | **63.16** | **56.25** | **75.68** | **49.82** | **41.56** | **56.87** |

**General Domain Classification.** To evaluate the versatility of *PMoE* in adapting to general visual tasks beyond the medical domain, we conducted experiments on two established benchmarks: FGVC and VTAB-1K. The results in Table 2 compare our approach with several recent methods, including VPT (Jia et al., 2022), GaPT (Yoo et al., 2023), and LSPT (Mo et al., 2024b). The FGVC benchmark consists of five fine-grained classification datasets, where small intra-class differences make adaptation particularly challenging. As shown in Table 2, the proposed *PMoE* demonstrates superior performance across multiple FGVC tasks, notably improving results on CUB, Flowers, and Cars datasets. Specifically, *PMoE* outperforms LSPT (Mo et al., 2024b) by 1.22 on the CUB dataset and 1.01 on Flowers. To further extend our assessment, we employed the VTAB-1K benchmark, which includes 19 diverse tasks across three categories: Natural, Specialized, and Structured. These tasks represent a variety of real-world challenges, providing a robust testing ground for adaptation methods. Our *PMoE* outperforms existing approaches in all three categories, achieving improvements of 0.69 on Natural, 0.63 on Specialized, and 0.59 on Structured tasks when compared to LSPT. These results validate the general applicability of *PMoE*, proving its adaptability across diverse general-domain tasks and establishing its competitiveness in fine-grained classification.

**Medical Domain Classification.** The evaluation of our proposed *PMoE* on medical imaging tasks, as summarized in Table 3, demonstrates its superior performance in handling various complex medical datasets, including tasks like polyp detection and skin analysis. In polyp detection tasks, such as Kvasir, *PMoE* outperforms state-of-the-art methods like LSPT (Mo et al., 2024b) by 4.15, and in skin lesion, we observe a 1.88 improvement. The results indicate that using Mixture-of-Experts significantly enhances the model's capacity to differentiate intricate patterns in medical images. Moreover, as detailed in Table 4, our *PMoE* provides significant improvements over existing methods for X-ray images, especially in distinguishing subtle features in chest and bone X-rays. In terms of OCT, CT, and MRI modalities, Table 5 highlights superior performance in modalities requiring high-detail orientation, such as brain tumor identification and chest CT analysis. These findings highlight the generality of the proposed *PMoE* in medical imaging tasks, showing significant improvements across a wide range of applications, from cell and polyp detection to complex tasks like skin lesion identification and ophthalmic analysis. By leveraging its mixture-of-experts framework, *PMoE* offers state-of-the-art performance with minimal additional parameters, making it a highly effective model for medical domain adaptation.

Table 4: Comparison results of visual prompt tuning of DINO v2 pre-trained vision transformers on X-ray images. Total Params denotes the total number of parameters for the backbone encoder ViT-B, prompt tokens or adapter parameters, and the task heads.

| Method | Total Params | Vindr Lung | CBIS Breast | COVIDx Lung | SYMH Shoulder | RSNAB Bone | CheXpert Chest | RSNA Lung |
|---|---|---|---|---|---|---|---|---|
| VPT (Jia et al., 2022) | 1.02X | 65.73 | 74.61 | 76.18 | 76.86 | 51.72 | 70.85 | 69.25 |
| VPT (Mo et al., 2024b) | 1.04X | 66.02 | 74.89 | 76.43 | 77.12 | 51.98 | 71.06 | 69.52 |
| VPT + *PMoE* (ours) | 1.04X | **68.31** | **76.75** | **78.56** | **79.38** | **54.23** | **73.58** | **72.81** |
| GaPT (Yoo et al., 2023) | 1.02X | 66.92 | 75.15 | 77.25 | 77.25 | 52.83 | 71.37 | 70.29 |
| GaPT (Yoo et al., 2023) | 1.04X | 67.37 | 75.52 | 77.83 | 77.79 | 53.46 | 71.85 | 70.78 |
| GaPT + *PMoE* (ours) | 1.04X | **69.82** | **77.85** | **80.16** | **80.07** | **56.52** | **74.69** | **74.15** |
| LSPT (Mo et al., 2024b) | 1.05X | 67.87 | 76.23 | 78.33 | 77.96 | 53.51 | 71.92 | 70.86 |
| LSPT (Mo et al., 2024b) | 1.10X | 68.59 | 76.92 | 78.98 | 78.73 | 54.37 | 72.68 | 71.62 |
| LSPT + *PMoE* (ours) | 1.10X | **73.21** | **80.53** | **82.39** | **82.26** | **58.79** | **76.85** | **76.91** |

Table 5: Comparison results of visual prompt tuning of DINO v2 pre-trained vision transformers on OCT, CT, and MRI images. Total Params denotes the total number of parameters for the backbone encoder ViT-B, prompt tokens or adapter parameters, and the task heads.

| Method | Total Params | Heide Eye | CC-CCII Chest | Mosmed Chest | COVID-C Chest | RICORD Chest | PPMI Brain | Tumor Brain |
|---|---|---|---|---|---|---|---|---|
| VPT (Jia et al., 2022) | 1.02X | 64.78 | 61.26 | 63.65 | 61.78 | 59.53 | 56.93 | 63.37 |
| VPT (Jia et al., 2022) | 1.04X | 65.02 | 61.53 | 63.89 | 62.03 | 59.82 | 57.26 | 63.69 |
| VPT + *PMoE* (ours) | 1.04X | **67.35** | **64.06** | **66.52** | **65.38** | **63.21** | **60.21** | **66.82** |
| GaPT (Yoo et al., 2023) | 1.02X | 65.06 | 61.37 | 63.69 | 61.95 | 59.71 | 56.97 | 63.52 |
| GaPT (Yoo et al., 2023) | 1.04X | 65.59 | 61.82 | 64.13 | 62.39 | 60.23 | 57.43 | 63.98 |
| GaPT + *PMoE* (ours) | 1.04X | **68.78** | **65.32** | **67.52** | **65.89** | **63.51** | **60.87** | **67.39** |
| LSPT (Mo et al., 2024b) | 1.05X | 65.23 | 61.56 | 63.75 | 62.12 | 59.85 | 57.08 | 63.67 |
| LSPT (Mo et al., 2024b) | 1.10X | 65.95 | 62.34 | 64.52 | 62.95 | 60.63 | 57.86 | 64.45 |
| LSPT + *PMoE* (ours) | 1.10X | **68.72** | **65.83** | **68.31** | **67.26** | **65.38** | **61.57** | **69.16** |

Table 6: Comparison results of visual prompt tuning of MAE & MoCo v3 pre-trained vision transformers on ADE-20K for semantic segmentation. SS and MS denote single-scale and multi-scale, respectively.

| Method | MAE | | MoCo v3 | |
|---|---|---|---|---|
| | mIoU (SS) | mIoU (MS) | mIoU (SS) | mIoU (MS) |
| VPT (Jia et al., 2022) | 37.76 | 38.80 | 35.50 | 37.15 |
| VPT + *PMoE* (ours) | **38.25** | **39.56** | **36.23** | **38.12** |
| GaPT (Yoo et al., 2023) | 38.44 | 39.81 | 36.81 | 38.55 |
| VPT-Deep + *PMoE* (ours) | **39.15** | **40.75** | **37.92** | **39.89** |
| LSPT (Mo et al., 2024b) | 39.72 | 41.51 | 37.92 | 39.73 |
| LSPT + *PMoE* (ours) | **40.68** | **42.87** | **39.36** | **41.58** |

**General Domain Segmentation.** In addition to classification tasks, we evaluate the effectiveness of our proposed *PMoE* on semantic segmentation. Following prior works (Jia et al., 2022; Yoo et al., 2023), we adopt the SETR-PUP (Zheng et al., 2021) model as the segmentation transformer framework on the ADE20K dataset (Zhou et al., 2017; 2018). Table 6 reports a comparative segmentation performance analysis between our *PMoE* and existing visual prompt tuning approaches. We utilize both MAE and MoCo v3 pre-trained ViT-B/16 weights, ensuring a comprehensive evaluation across multiple pre-trained models. Compared to VPT (Jia et al., 2022) and GaPT (Yoo et al., 2023), our *PMoE* demonstrates significant performance gains across all key segmentation metrics. For both MAE and MoCo v3 backbones, we observe substantial improvements of up to 1.36 in MAE and 1.85 in MoCo v3, showcasing the robust adaptability of *PMoE* for dense prediction tasks. These improvements can be attributed to the inherent structure of our mixture-of-experts approach, which effectively leverages multiple experts to fine-tune visual representations, resulting in superior segmentation.

**Medical Domain Segmentation.** In the medical domain, segmentation plays a crucial role in identifying and delineating anatomical structures or pathological regions. We benchmarked our *PMoE* against several state-of-the-art visual prompt tuning methods on diverse medical segmentation datasets. Table 7 provides a detailed comparison of the performance across key medical segmentation tasks, including Kvasir polyp and Skin lesion segmentation. Our *PMoE* significantly outperforms prior approaches (Jia et al., 2022; Yoo et al., 2023) in all metrics, particularly in complex segmentation tasks

Table 7: Comparison results of visual prompt tuning of MAE & MoCo v3 pre-trained vision transformers on Kvasir Polyp and Skin Lesion for medical segmentation. mIoU metrics on a single scale are reported.

| Method | MAE | | MoCo v3 | |
|---|---|---|---|---|
| | mIoU (Kvasir-seg) | mIoU (Skin) | mIoU (Kvasir-seg) | mIoU (Skin) |
| VPT (Jia et al., 2022) | 42.87 | 74.23 | 40.65 | 72.68 |
| VPT + *PMoE* (ours) | **43.52** | **75.16** | **42.08** | **73.76** |
| GaPT (Yoo et al., 2023) | 43.95 | 75.52 | 42.17 | 73.89 |
| VPT-Deep + *PMoE* (ours) | **45.38** | **77.21** | **43.79** | **75.86** |
| LSPT (Mo et al., 2024b) | 45.81 | 77.63 | 43.92 | 76.27 |
| LSPT + *PMoE* (ours) | **47.95** | **80.35** | **46.56** | **79.83** |

Table 8: Ablation studies on Expert Prompt Tokens (EPTs) and Dispatcher.

| EPTs | Dispatcher | CUB | Flowers | Cars | Dogs | NABirds |
|---|---|---|---|---|---|---|
| ✗ | ✗ | 82.95 | 94.65 | 79.37 | 83.62 | 76.21 |
| ✓ | ✗ | 83.07 | 94.71 | 79.45 | 83.69 | 76.32 |
| ✓ | ✓ | **83.38** | **94.85** | **79.72** | **83.95** | **76.73** |

Table 9: Ablation studies on the type of mixture-of-experts models.

| Expert 1 | Expert 2 | CUB | Flowers | Cars | Dogs | NABirds |
|---|---|---|---|---|---|---|
| MoCo v3 | CLIP | 83.38 | 94.85 | 79.72 | 83.95 | 76.73 |
| DINO | CLIP | **83.76** | **95.27** | **80.52** | **84.65** | **77.38** |
| DINO | MoCo v3 | 83.17 | 94.72 | 79.56 | 83.78 | 76.58 |
| DINO | MAE | 83.28 | 94.86 | 79.67 | 83.89 | 76.67 |
| MAE | CLIP | 83.56 | 95.01 | 80.06 | 84.26 | 76.95 |

Table 10: Comparison results on ViT-L/16 weights.

| Method | Total Params | Natural | Specialized | Structured |
|---|---|---|---|---|
| VPT (Jia et al., 2022) | 1.03X | 70.86 | 83.76 | 43.05 |
| VPT (Jia et al., 2022) | 1.06X | 71.93 | 84.98 | 44.12 |
| VPT + *PMoE* (ours) | 1.06X | **72.87** | **85.97** | **45.06** |
| GaPT (Yoo et al., 2023) | 1.03X | 75.42 | 83.97 | 49.78 |
| GaPT (Yoo et al., 2023) | 1.06X | 76.67 | 85.06 | 50.86 |
| GaPT + *PMoE* (ours) | 1.06X | **77.73** | **86.19** | **51.97** |
| LSPT (Mo et al., 2024b) | 1.06X | 78.68 | 86.75 | 53.86 |
| LSPT (Mo et al., 2024b) | 1.12X | 80.25 | 87.98 | 55.03 |
| LSPT + *PMoE* (ours) | 1.12X | **81.67** | **89.12** | **56.35** |

that require high spatial precision. For instance, we observe a 2.14 improvement in Kvasir polyp and a 2.72 boost in lesion when compared to LSPT (Mo et al., 2024b). The improved performance of our *PMoE* in medical segmentation tasks is attributed to its MoE-based architecture, which allows for the specialization of different experts, each focusing on specific segmentation challenges. These results underscore the potential of our *PMoE* as a state-of-the-art solution for medical image segmentation.

## 4.3 EXPERIMENTAL ANALYSIS

In this section, we performed ablation studies to demonstrate the benefit of Expert Prompt Tokens and Dispatcher. We also conducted extensive experiments to explore the impact of pre-trained model types, model size, and number of prompt layers.

**Expert Prompt Tokens & Dispatcher.** To validate the efficacy of the Expert Prompt Tokens (EPTs) and Dispatcher components in our prompt tuning framework, we conduct ablation studies, isolating the impact of each component. Table 8 reports the results of experiments comparing the full pMoE model with variants that exclude either Expert Prompt Tokens or Dispatcher. We observe a notable drop in performance when either component is removed, highlighting their synergistic role in improving task adaptation and performance across various benchmarks. Our analysis shows that the inclusion of Expert Prompt Tokens significantly enhances the model's ability to specialize prompts based on task complexity. Similarly, Dispatcher helps in distributing task-specific knowledge across deeper layers of the network, allowing for better contextual understanding and representation. These elements lead to a consistent boost across all tasks, confirming their importance in our design.

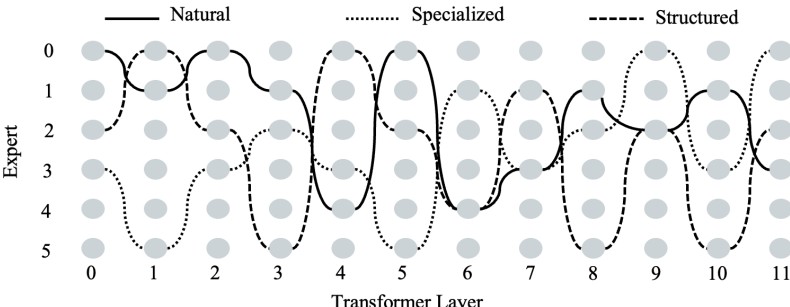

Figure 2: Visualization of Mixture-of-Experts path. Our *pMoE* can dynamically choose a distinct, task-specific path of experts for each benchmark type, demonstrating the ability to adapt to the particularities of each task.

**Pre-trained Model Types.** The choice of pre-trained model types plays a crucial role in downstream task performance. We conduct experiments using four different types of pre-trained models: MAE, CLIP, DINO, and MoCo v3. Table 9 compares the results across these models. As expected, models pre-trained with contrastive learning objectives such as CLIP and MoCo v3 show improved generalization in visual domains, while MAE and DINO offer strong feature representations for image classification. Our findings indicate that the pMoE framework is highly adaptable to various pre-training regimes, consistently outperforming baseline methods regardless of the underlying model. However, the highest improvements are seen when utilizing DINO and MoCo v3, suggesting that these pre-trained models offer richer representations, which benefit from the specialized learning pathways provided by MoE.

**Large Pre-trained Backbone.** We also evaluate the effect of using larger backbone models in conjunction with the *pMoE* framework. Table 10 compares results when scaling up from ViT-B to ViT-L models. As expected, larger backbone models improve performance across most tasks due to their increased capacity to capture finer details. However, the pMoE framework consistently demonstrates its capacity to improve over baseline methods, regardless of the model size. This scalability highlights the versatility of our method, making it applicable to both standard and large-scale models.

**Visualization of Mixture-of-Experts Path.** To illustrate the adaptability of *pMoE*, we visualize the expert selection paths with the maximum activation value on experts across different layers for various types of benchmarks: Natural, Specialized, and Structured. As shown in Figure 2, *pMoE* dynamically selects a distinct, task-specific path of experts for each benchmark type, demonstrating the model's ability to adapt to the particularities of each task. This adaptive mechanism allows *pMoE* to efficiently route information through the most relevant experts at different layers, enabling the model to focus on task-specific features while preserving generalization. The visualizations highlight the diverse paths chosen for different tasks, reflecting how the framework optimally allocates resources across layers based on the task complexity and domain, ultimately contributing to the superior performance observed in our experiments.

## 5 CONCLUSION

In this work, we present *pMoE*, a novel prompt-tuning framework that leverages Mixture-of-Experts mechanisms to improve visual adaptation across a wide range of tasks. By integrating Expert Prompt Tokens and Dispatcher, our approach achieves significant gains in performance compared to existing visual prompt tuning techniques. Our ablation studies clearly demonstrate the distinct benefits brought by each of these components, illustrating how *pMoE* dynamically allocates model capacity based on task complexity, leading to more efficient and accurate visual adaptation. Extensive experiments across multiple domains validate the versatility and effectiveness of our method. On general domain tasks, *pMoE* shows consistent improvements in fine-grained classification and semantic segmentation, outperforming traditional techniques on diverse benchmarks. In the medical domain, our *pMoE* demonstrates remarkable advancements in both classification and segmentation tasks. Our exploration of different pre-trained model types further confirms the robustness of our approach, highlighting the generalizability of *pMoE* across various vision transformers and initialization methods. Moreover, we have shown that scaling to larger backbone architectures significantly enhances the model's ability.

ETHICS STATEMENT

We are committed to adhering to the ICLR Code of Ethics. Our research primarily utilizes publicly available datasets, ensuring transparency and accessibility for the broader research community. While our experiments are conducted on widely used datasets from both general and medical domains, we acknowledge the importance of responsible AI application, particularly in sensitive areas like healthcare. We encourage the community to apply our methods ethically and consider potential biases in real-world data that could impact the fairness and accuracy of models in deployment.

REPRODUCIBILITY STATEMENT

To ensure the reproducibility of our results, we have provided detailed explanations of our methodology and experimental setups in Section 4. Our ablation studies and algorithmic design are thoroughly documented, and additional experimental details are included in the Appendices A and B. We are committed to sharing our code and pre-trained models with the research community upon publication, allowing for transparency, easy replication of our experiments, and further development.

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

APPENDIX

In this appendix, we present addition implementation and datasets details in Section A. We also present the detailed algorithm for our *PMoE* in Section B. We further provide additional experimental analyses in Section C.

## A    EXPERIMENTAL DETAILS

**Datasets.** For the general domain, we leverage two popular classification benchmarks: FGVC (Wah et al., 2011; Nilsback & Zisserman, 2008; Gebru et al., 2017; Khosla et al., 2011; Van Horn et al., 2015) and VTAB-1K (Zhai et al., 2019). The FGVC benchmark consists of five fine-grained classification tasks, including CUB-200-2011 (Wah et al., 2011), Oxford Flowers (Nilsback & Zisserman, 2008), Stanford Cars (Gebru et al., 2017), Stanford Dogs (Khosla et al., 2011), and NABirds (Van Horn et al., 2015). We follow the same training and validation split as prior work (Jia et al., 2022; Yoo et al., 2023; Mo et al., 2024b). For VTAB-1K (Zhai et al., 2019), we include 19 diverse visual classification tasks grouped into three categories: Natural images, Specialized images captured with specific equipment, and Structured images for object counting. Each task contains 1000 training samples, and we follow the standard splits used in previous work (Jia et al., 2022; Yoo et al., 2023; Mo et al., 2024b). For the medical domain, we use a broad set of datasets from the Med-VTAB benchmark (Mo et al., 2024a), covering a range of medical tasks. These include: Color medical images include nine datasets, including HyperKvasir (Borgli et al., 2020), MESAD Prostatectomy (Bawa et al., 2021), Kvasir (Kvasirv2), AMLC (Matek et al., 2019), LHNCBC (Lhncbc malaria), MLLBone (Matek et al., 2021), APTOS (Aptos 2019 blindness detection), EyePACS (Kaggle dr dataset (eyepacs)), and ISIC (Skin lesion images for melanoma classification). X-ray images consist of seven datasets, including Vindr (Nguyen et al., 2022), COVIDx (Wang et al., 2020), RSNA (Shih et al., 2019), CBIS (Lee et al., 2017), SYMH (Shoulder X-ray Classification), RSNA Bone (Halabi et al., 2019), and CheXpert (Irvin et al., 2019). OCT, CT, and MRI modalities contains seven datasets, including Heidelberg (Kermany et al., 2018), CC-CCII (Zhang et al., 2020b), Mosmed (Morozov et al., 2020), COVID-C (Rahimzadeh et al., 2021), RICORD (Tsai et al., 2021), PPMI (Marek et al., 2011), and Brain-Tumor (Brain Tumor MRI Dataset). For the segmentation tasks, we include the ADE20K (Zhou et al., 2017; 2018), Kvasir-seg polyp (Jha et al., 2020) and the ISIC skin lesion dataset (Gutman et al., 2016). The ADE20K dataset consists of 20K images fully annotated with objects, spanning over 150 semantic categories. The Kvasir-seg dataset consists of 1000 polyp images and their corresponding binary label masks, while the ISIC skin lesion dataset includes 900 training and 379 test images for the task of dermoscopic image segmentation.

**Implementation.** We implement our *PMoE* framework using PyTorch (Paszke et al., 2019). All experiments are conducted on NVIDIA A100 GPUs, with 80 GB of memory, allowing us to efficiently fine-tune models across diverse datasets. For our experiments, we use Vision Transformer (ViT-B/16) models pre-trained on ImageNet-21K (Deng et al., 2009). The pre-trained models are initialized with publicly available weights and are then adapted for downstream tasks using the proposed *PMoE* framework. We fine-tune the models using the AdamW optimizer (Loshchilov & Hutter, 2017), with a learning rate of 1e-4 and weight decay of 1e-5. The batch size is set to 32 across all datasets, and models are trained for 30 epochs for each experiment. For FGVC datasets, such as CUB, Flowers, and NABirds, we follow the standard training and validation splits provided in the respective datasets. For VTAB-1K benchmarks, we use the splits from prior works (Jia et al., 2022; Yoo et al., 2023; Mo et al., 2024b). While *PMoE* introduces additional complexity through the use of multiple experts and dynamic dispatching, the increase in FLOPs (floating-point operations) is kept minimal due to the sparse activation of experts. In practice, we observe that the computational overhead remains manageable, and the trade-off is justified by the significant performance gains achieved in both general and medical visual adaptation tasks.

For the ablation studies on prompt layers (Table 13), we vary the number of layers that receive prompt tokens. Specifically, we experiment with using 3, 6, 9, and 12 prompt layers. As demonstrated in the results, adding more prompt layers improves the model's ability to capture complex visual features, but with diminishing returns beyond 9 layers due to increased computational overhead. For each experiment, we vary the number of experts and prompt layers, as shown in Tables 12 and 13. As the number of experts increases, the model captures richer domain-specific knowledge, leading to improved performance, particularly on complex datasets such as FGVC's fine-grained classification

---

**Algorithm 1** *PMoE*: Mixture-of-Experts Prompt Tuning Framework

---

**Input:** Image $\mathbf{X}$, $K$ pre-trained experts, and each one includes one patchifier (i.e., $\text{Patchifier}_{\text{expert}_k}$) and $L$ transformer layers (i.e., $\text{TransLayer}^l_{\text{expert}_k}$ for the $l$-th one).

**Introduced learnable weights:** Expert Prompt Tokens (EPTs) $\mathbf{P}^l = \{\mathbf{P}^l_{\text{expert}_1}, \dots, \mathbf{P}^l_{\text{expert}_K}\}$, where $\mathbf{P}^l_{\text{expert}_k} \in \mathbb{R}^{N_p \times D}$ and $D$ is the dimension of all kinds of tokens, MLPs in the dispatcher layer with input dimension $D$ and output dimension $K$.

**Output:** Final task-specific output $\mathbf{y}$.

1: Extract patch tokens $\mathbf{Z}^0_{\text{expert}_k} \in \mathbb{R}^{N_z \times D}$ from input image $\mathbf{X}$ for each expert $k = 1, \dots, K$:

$$\mathbf{Z}^0_{\text{expert}_k} = \text{Patchifier}_{\text{expert}_k}(\mathbf{X})$$

2: **for** $l = 0$ to $L - 1$ **do**
3:     **for** $k = 1$ to $K$ **do**
4:         Compute dispatching weights $\mathbf{D}^l_{\text{expert}_k} \in \mathbb{R}^{N_p \times K}$ for the $k$-th expert from the newly added EPTs $\mathbf{P}^l_{\text{expert}_k}$, accumulated prompts $\mathbf{Z}^l_{P,\text{expert}_k}$ (excluded for the first layer), and patch tokens $\mathbf{Z}^l_{\text{expert}_k}$:

$$[\mathbf{D}^l_{E,\text{expert}_k}; \mathbf{D}^l_{P,\text{expert}_k}; \mathbf{D}^l_{Z,\text{expert}_k}] = \text{MLPs}([\mathbf{P}^l_{\text{expert}_k}; \mathbf{Z}^l_{P,\text{expert}_k}; \mathbf{Z}^l_{\text{expert}_k}])$$

$$\mathbf{D}^l_{\text{expert}_k} = \text{softmax}(\mathbf{D}^l_{E,\text{expert}_k} + \mathbf{D}^l_{P,\text{expert}_k} + \text{Average}(\mathbf{D}^l_{Z,\text{expert}_k}, \text{d} = 1), \text{d} = 2)$$

5:         Fuse tokens $\mathbf{P}^l \in \mathbb{R}^{K \times N_p \times D}$ using dispatching weights:

$$\hat{\mathbf{P}}^l_{\text{expert}_k}[n] = \mathbf{D}^l_{\text{expert}_k}[n] \times \mathbf{P}^l[\cdot, n]$$

$$\hat{\mathbf{P}}^l_{\text{expert}_k} = [\hat{\mathbf{P}}^l_{\text{expert}_k}[1], \dots, \hat{\mathbf{P}}^l_{\text{expert}_k}[n], \dots, \hat{\mathbf{P}}^l_{\text{expert}_k}[N_p]]$$

6:         Prepend expert prompt tokens to patch tokens to the $l$-th transformer layer:

$$[\mathbf{Z}^{l+1}_{P,\text{expert}_k}; \mathbf{Z}^{l+1}_{\text{expert}_k}] = \text{TransLayer}^{l+1}_{\text{expert}_k}([\hat{\mathbf{P}}^l_{\text{expert}_k}; \mathbf{Z}^l_{\text{expert}_k}])$$

7:     **end for**
8: **end for**
9: Combine outputs from all experts $\{\mathbf{Z}^L_{\text{expert}_1}, \dots, \mathbf{Z}^L_{\text{expert}_K}\}$ into a unified representation:

$$\mathbf{Z}_{\text{combined}} = \text{Average}(\mathbf{Z}^L_{\text{expert}_1}, \dots, \mathbf{Z}^L_{\text{expert}_K})$$

10: Pass the combined representation through a task-specific head to produce the output:

$$\mathbf{y} = \text{Head}(\mathbf{Z}_{\text{combined}})$$

11: **Return:** Final task-specific output $\mathbf{y}$.

---

tasks. For classification tasks, we report accuracy as the primary metric, including the mean accuracy across tasks in VTAB-1K and FGVC. For segmentation tasks, Intersection over Union (IoU) is used as the evaluation metric. Each experiment is run multiple times, and we report the average performance across multiple runs to ensure robustness and consistency of results.

## B   Algorithm for *PMoE*

In this section, we describe the overall procedure for our proposed framework *PMoE*, which leverages a dynamic Mixture-of-Experts (MoE) prompt tuning mechanism to integrate knowledge from multiple domain experts. The key components of the algorithm include the injection of Expert Prompt Tokens (EPTs) and the dynamic dispatching mechanism, which ensures efficient use of expert knowledge, as shown in Algorithm 1.

**Designs for the Dispatcher Layer.** Considering the computation efficiency, we first provide an initial simple implementation for the dispatcher layer, which is MLPs. Experiment results in Section C demonstrate that simple MLPs already show competitive performance. We notice that improvements

Table 11: Comparison results of visual prompt tuning shallow models (VPT-Shallow) on VTAB-1K benchmarks. Numbers in (·) denote the number of downstream datasets.

| Method | Total Params | Natural (7) | Specialized (4) | Structured (8) | Average |
|---|---|---|---|---|---|
| VPT-Shallow (Jia et al., 2022) | 1.01X | 67.34 | 82.26 | 37.55 | 57.94 |
| VPT-Shallow (Jia et al., 2022) | 1.02X | 67.51 | 82.39 | 37.72 | 58.10 |
| VPT-Shallow + *pMoE* (ours) | 1.02X | **68.72** | **83.81** | **39.58** | **59.63** |

Table 12: Ablation studies on the number of experts in MoE-MLPs for the dispatcher.

| # expert | CUB | Flowers | Cars | Dogs | NABirds |
|---|---|---|---|---|---|
| 3 | 83.17 | 94.76 | 79.51 | 83.75 | 76.45 |
| 6 | 83.38 | 94.85 | 79.72 | 83.95 | 76.73 |
| 9 | **83.43** | **94.87** | **79.75** | **83.98** | **76.78** |

regarding the dispatcher layer can be further made here. To this end, we find a better design for the dispatcher layer, having superior performance without introducing much more computation costs at the same time. We leverage the idea of MoE once again. Specifically, each row in $\mathbf{D}$ represents a decision for dispatching tokens, with each decision learned by a set of MLP parameters. Here, each MLP making a decision can be abstracted as a dispatching expert, with each expert producing an integrated token. Innovatively, we introduce multiple experts and a router during the training process to determine which dispatching expert is activated. This approach allows for more flexible and dynamic decision-making, while the sparse activation of experts does not impose a significant computational burden.

## C  EXPERIMENTAL ANALYSIS

In this section, we provide a detailed experimental analysis to validate the effectiveness of *pMoE*. We perform ablation studies to explore the impact of key design choices, such as the use of shallow models, the number of experts, and the number of prompt layers. These studies are critical to understanding how different configurations of *pMoE* affect its generalization across diverse tasks.

**VPT-Shallow Models.** We analyze the impact of applying visual prompt tuning to shallow models by reducing the number of prompt layers. As shown in Table 11, the VPT-Shallow (Jia et al., 2022) models perform reasonably well on simpler tasks, particularly in the Natural and Specialized categories. However, there is a noticeable drop in performance on more complex tasks, such as those in the Structured category. Specifically, while the VPT-Shallow method achieves an average score of 58.10, the Structured tasks are particularly challenging, with scores as low as 37.72. By integrating our proposed *pMoE* framework (VPT-Shallow + pMoE), we observe significant improvements across all categories, with a notable gain in the Structured category (+1.86) and a boost in the overall average (+1.53). These results highlight that while VPT-Shallow models are effective for less complex tasks, the addition of Mixture-of-Experts (MoE) mechanisms in pMoE allows for better handling of task complexity, ultimately leading to stronger overall performance across diverse tasks.

**Number of Experts.** Table 12 reports the results when varying the number of experts used in *pMoE*. As the number of experts increases, the model demonstrates improved performance across various datasets, particularly in complex tasks like CUB, Flowers, and NABirds. For instance, moving from 3 experts to 6 experts provides a noticeable boost in performance for most tasks, with the performance on CUB increasing from 83.17 to 83.38 and on Dogs from 83.75 to 83.95. However, the improvement begins to plateau as the number of experts increases beyond 6. Moving from 6 experts to 9 shows only marginal gains, with slight improvements in CUB (+0.05) and NABirds (+0.05), while the performance on Flowers and Cars remains nearly identical. This suggests that while increasing the number of experts allows the model to capture more diverse domain-specific knowledge, there are diminishing returns beyond a certain point. Adding too many experts can increase computational costs without providing proportional gains in performance.

**Number of Prompt Layers.** Table 13 explores the impact of varying the number of prompt layers in *pMoE*. The results show that increasing the number of prompt layers generally leads to better performance across most tasks. For example, performance on the CUB dataset improves from 83.08

Table 13: Ablation studies on the number of prompt layers.

| # layer | CUB | Flowers | Cars | Dogs | NABirds |
|---|---|---|---|---|---|
| 3 | 83.08 | 94.68 | 79.42 | 83.68 | 76.29 |
| 6 | 83.15 | 94.72 | 79.53 | 83.79 | 76.43 |
| 9 | 83.29 | 94.78 | 79.65 | 83.86 | 76.62 |
| 12 | **83.38** | **94.85** | **79.72** | **83.95** | **76.73** |

with 3 layers to 83.38 with 12 layers. Similarly, in the NABirds dataset, the score increases from 76.29 to 76.73. These improvements indicate that more prompt layers enable the model to capture more complex and nuanced features, leading to better adaptation for fine-grained classification tasks. However, there is a point of diminishing returns. For instance, moving from 9 layers to 12 layers provides only marginal gains, with increases of just 0.09 on CUB and 0.11 on NABirds. Additionally, increasing the number of prompt layers introduces more computational overhead, which could affect model efficiency. Therefore, while using more prompt layers can enhance model performance, it is essential to balance the number of layers with computational costs to achieve the best trade-off between performance and efficiency.

