# OpenReview forum: "pMoE: Prompting Diverse Experts Together Wins More in Visual Adaptation"
_ICLR.cc/2025/Conference — ICLR 2025 Poster_

### Official Review · Reviewer_eefg · 2024-10-26

**Soundness:** 2
**Presentation:** 3
**Contribution:** 2
**Rating:** 5
**Confidence:** 5

**Summary:**

The paper introduces pMoE (Mixture-of-Experts prompt tuning), an innovative prompt tuning approach that enhances parameter-efficient fine-tuning for visual adaptation tasks like classification and segmentation. Traditional prompt tuning methods use knowledge from a single pre-trained model, either general or specialized (e.g., medical), which limits the model’s adaptability across diverse tasks. pMoE addresses this by integrating multiple domain experts within a single framework.

**Strengths:**

1. the method is novel and has practical significance
2. paper did a comprehensive evaluation
3. the paper is straightforward and easy to follow

**Weaknesses:**

1. the tested datasets are mostly simple, for example ISIC is not a challenging dataset, and thus the performance gaining may mainly come from over-fitting.
2. the experiments can not well back the claim that combining multiple experts have sufficient advantage over training one foundation model.
3. the authors have not done enough analysis to explore the multi-expert integration process in-depth.

**Questions:**

see weakness

---

### Official Review · Reviewer_wRQ8 · 2024-10-29

**Soundness:** 4
**Presentation:** 3
**Contribution:** 4
**Rating:** 8
**Confidence:** 3

**Summary:**

This paper propose a novel framework prompt tuning called Mixture-of-Experts Prompt Tuning (pMoE), which leveraging knowledge from
multiple expert domains. This method help to enhances the model’s versatility and applicability to a broad spectrum of tasks. They first introduce expert-specific prompt tokens to each pre-trained expert. Next, they introduce learnable dispatcher modules that can dynamically select and fuse tokens from diverse experts for each layer having prompt. They conduct experiments on 47 visual adaptation benchmarks, including both classification and segmentation tasks from general and medical domains and achieve state-of-the-art performance while strikes a balance between computational efficiency and adaptation efficacy.

**Strengths:**

- Proposed a novel prompt tuning framework that can unify diverse domain-specific models, enhances the model’s versatility with computational efficiency.
- Extensive experiments were conducted and had good results.
- The paper is well-written and easy to follow.

**Weaknesses:**

- I see that this prompt learning method is trained on A100-80GB GPUs, which I am afraid of being not available in many places. Can this method be trained on smaller systems ?

**Questions:**

- Can you give me the detail about approximate total time to train the models using your method and the FLOPs of your method.

---

### Official Review · Reviewer_Q1EU · 2024-10-30

**Soundness:** 2
**Presentation:** 2
**Contribution:** 2
**Rating:** 6
**Confidence:** 3

**Summary:**

This paper proposes a novel Mixture-of-Experts prompt tuning method called pMoE, which effectively combines knowledge from multiple domain experts in a unified model framework through expert-specific prompt tokens and a learnable dispatcher. pMoE introduces expert-specific prompt tokens at various prompt layers and utilizes a dynamic token dispatching mechanism to optimize the contribution of each domain expert during the adaptation process. By integrating domain knowledge from diverse experts, pMoE significantly enhances the model's versatility and applicability to a broad spectrum of tasks. Extensive experiments on 47 adaptation tasks, including classification and segmentation in both general and medical domains, demonstrate that pMoE not only achieves superior performance with significant improvements but also offers an optimal trade-off between computational efficiency and adaptation effectiveness compared to existing methods.

**Strengths:**

1. Proposed pMoE that effectively integrates knowledge from multiple domain experts using expert-specific prompt tokens and a dynamic dispatcher.
2. Demonstrated strong results in both general and medical domains, showing the method's adaptability across different fields and task types.

**Weaknesses:**

1. While the results are comprehensive, the paper lacks visualizations that could help explain the inner workings of the **Dynamic Dispatcher**. For example, a more detailed breakdown of how the dispatcher allocates weights across different experts in specific tasks would make the method’s dynamics clearer.
2. Although the paper performs ablation studies on the number of experts (as shown in Table 12), it would be beneficial to see a more in-depth analysis of **how different domains influence each other** when multiple experts are involved. This could provide insights into how domain-specific knowledge is shared or conflicts across tasks.
3. The paper compares pMoE with several state-of-the-art methods, but some key baselines are missing, particularly methods that utilize **multi-model fusion** or other advanced **transfer learning** techniques. Including these would provide a more robust comparison.

**Questions:**

1. What is the specific architecture of the dispatcher layer? The dispatcher plays a critical role in pMoE, but the paper does not provide detailed information about the design and implementation of the dispatcher layer. How does it effectively utilize the prompt tokens from different experts?
2. Why choose expert-specific prompt tokens instead of shared prompts? What are the advantages of using expert-specific prompt tokens rather than shared prompt tokens? How does this choice affect the model's performance and parameter efficiency?
3. How does pMoE handle possible conflicts or redundancies among experts? When there are conflicts or overlaps in the domain knowledge of multiple experts, how does pMoE effectively coordinate and utilize this information?
4. The paper evaluates pMoE using various pre-trained models (e.g., DINO, MoCo v3, MAE). Given that different pre-training methods lead to different feature representations, how does the choice of pre-trained model affect the performance of pMoE? Could the authors elaborate on any observed patterns regarding which pre-training methods work best with pMoE?
I would be pleased if the author could consider raising my score after addressing my concerns.

---

> ### Comment · Reviewer_Q1EU · 2024-11-28
>
> Thank you very much to the authors for taking my suggestions into account. The revised version and the additional experiments have addressed most of my concerns. The extra experiments clearly demonstrate the dynamic contributions of different experts across various tasks. I sincerely appreciate the authors' efforts. As promised, I have raised my score to 6.

---

### Official Review · Reviewer_eZPn · 2024-11-04

**Soundness:** 2
**Presentation:** 2
**Contribution:** 2
**Rating:** 6
**Confidence:** 4

**Summary:**

The paper contributes a new prompt-tuning framework to adapt pre-trained models to new downstream tasks using a mixture of expert methods. For this, authors utilize a dynamic token dispatching strategy at different prompt layers to optimize the contribution of each domain expert. They conducted a wide range of experiments on both general and medical domains, including classification, segmentation, etc, and demonstrated improved results overall.

**Strengths:**

**Methodology**:
- Ideas of using a mixture of experts to combine different kinds of prompt tuning are interesting ideas.
- The author also proposes using dynamic gating to selectively choose and integrate tokens from various experts, which is essential to (1) allow communication among experts for better information exchange and (2) determine which information should be retained.

**Experiments**:
- Reviewers appreciate the efforts of the paper to conduct several experiments and compare with several relevant models. Ablation studies on Expert Prompt Tokens and Dispatcher type, etc.
- Figure 2 illustrates how the mixture-of-expert path is interesting in understanding how the model likely works in practice.

**Weaknesses:**

The reviewer found the following major weaknesses in this paper:

**Methodology**:
- The writing part for Section 3.3 is very convoluted and is difficult to grasp ideas beyond. For e.g., the following sentence is *These tokens are the added EPTs for all experts $P^{l}$, accumulated prompts from the last layer $Z_{P, expert_{k}}^{l}$, and patch tokens of the current expert $Z_{expert_{k}}^{l}$*. So the question is, what is **these** here? and what is exactly the equation for using $\hat{P}_{expert-k}^{l}$? Will it be used to define the matrix $P^{l+1}$ for the next layer?

- Another important question is what the Expert K represents. Does it mean the k-th domain expert is a specific pre-trained model? (like what authors write below equation (3) with DINO and LVM-Med), or each k-th domain expert is just another prompt vector?

- Figure 1 is also confusing. For example, Reviewers don't understand the block output of Dispatcher Layer l, which has several rectangles and colors that are not defined in the notion. Furthermore, why do we have the same notations for the two first rectangles for Expert $k$, Expert $1$, and Expert $K$, which contradicts the equation (5)?

- The authors mentioned the *Mixture of Expert Prompt Tokens* and *Mixture of Experts Prompt Layers*. While the first term is presented in the paper, there is no discussion about *Mixture of Experts Prompt Layers*.

In summary, the lack of details and unclear presentation in Section 3.3 about how the *dispatch* and the meaning of *expert* make the Reviewer unable to follow and understand contributions in experiments.

**Experiments**:

- The first and also the most important questions authors did not provide in their experiments are *how many experts are used in their settings*, i.e., the number of $k$? and *what are they (in terms of architectures)?* It is crucial to understand and analyze the results in the reported table.

- For instance, in the Medical Domain Segmentation task (line 416), authors follow the setting of [1], which integrated two ViT pre-trained models, one from the general domain and the other from the medical one. Given this, the question is how do authors adapt their method in this case with two pre-trained models? This question is indeed relevant to the above question (methodology) about the meaning of $k$-th domain expert. Do you choose K=2 for this case, or does each pre-trained ViT have different K-experts?

- **Repeat the same baselines with different results**: In particular., in Tables 2, 3, 4, and 5, why do baselines like VPT, GaPT, and LSPT repeat two times, but their total params and other results are different? The reviewer could not find experimental description information for these results, which raises a concern about the accuracy of baseline results.

- Another problem with Table 9 in Ablation studies. There are two columns *Model 1* and *Model 2* in the Table. Does that mean that the first expert is *Model 1* and the second expert is *Model 2*, or k = 2?

[1] Shentong Mo, Xufang Luo, Yansen Wang, and Dongsheng Li. A large-scale medical visual task adaptation benchmark, Arxiv 2024.

**Questions:**

As mentioned above, the Reviewer doesn't understand the right setting of the expert-k (and how you apply it to a single model and different models) and how dispatch learning works, which causes disconnections in most experiment parts. So can you please provide a details explanation for those questions as well as some other things in experiments (repeat the same baseline with different numbers)?

---

### Meta-Review · Area_Chair_vrDL · 2024-12-22

**Metareview:**

The paper introduces Mixture-of-Experts Prompt Tuning (pMoE), a novel framework for parameter-efficient fine-tuning that integrates multiple domain experts, enhancing adaptability across visual tasks. Reviewers praise its innovative methodology, comprehensive experiments on 47 benchmarks, strong performance, versatile utilities, and clear presentation. However, the paper lacks in-depth analysis of the multi-expert integration process and the dynamics of expert interactions. The evaluation is conducted on relatively simple datasets, raising concerns about the generalizability of the results, and there is insufficient comparison with other advanced transfer learning techniques. Most of these limitations have been addressed in the rebuttal. Overall, the method is innovative, and the paper is well-written, with the potential for further improvement and a more thorough exploration of the method's advantages.

**Additional Comments On Reviewer Discussion:**

During the discussion phase, reviewers raised concerns about the reliance on simpler datasets, insufficient evidence demonstrating the advantages of multi-expert models over single models, a lack of detailed analysis of the dispatcher mechanism, and ambiguities in the experimental setups. Most of these concerns were adequately addressed during the discussion stage, resulting in three positive scores after the rebuttal. The only negative score came from a reviewer who did not provide further feedback. Overall, while there is room for improvement, the paper is widely recognized as a valuable and novel contribution.

---

### Decision · Program_Chairs · 2025-01-22

Accept (Poster)